# Current Trends in Music Performance Anxiety Intervention

**DOI:** 10.3390/bs13090720

**Published:** 2023-08-29

**Authors:** Belén Gómez-López, Roberto Sánchez-Cabrero

**Affiliations:** Interfacultative Department of Evolutionary Psychology and Education, Faculty of Teacher Training and Education, Autonomous University of Madrid, 28049 Madrid, Spain; belen.gomez@estudiante.uam.es

**Keywords:** music performance anxiety, professional musicians, performance anxiety, PRISMA method, music performance

## Abstract

Music performance anxiety (MPA) is a natural, emotional, and physiological response to the stress of public performance. Debilitating forms of MPA are severe and persistent reactions that go beyond the normal adaptive response to music evaluation situations and can negatively impact the quality of musical performance and the musician’s life in general. Today, it affects numerous professional performers and can result in an inability to practice their profession, posing a significant threat to their professional activity. Despite its scope, studies exploring this issue and contributing to its resolution are scarce. Thus, this review aims to compile the significant advancements made in the last five years (2018–2023) in the treatment of MPA from a scientific perspective. For this purpose, the PRISMA method was used based on the results obtained from the Web of Science, Scopus, and Google Scholar databases. Reviewed are 10 studies that have made valuable contributions to this matter in this time frame after applying the quality filters using the PRISMA method. It is concluded that, although there are methodological shortcomings and sample limitations in the current research, this field registers advancements that provide valuable information to prevent or solve this problem in professional or aspiring musicians.

## 1. Introduction

Among specific anxieties, performance anxiety is considered by the American Psychiatric Association [1] as a category within social anxiety. More specifically, music performance anxiety (hereinafter MPA) encompasses a set of cognitive, physiological, and behavioral symptoms that impact the population of musicians when exposed to public musical performance [2]. The debilitating forms of this type of anxiety referred to in this study are distinguished from the symptoms of adaptive MPA by their greater intensity and persistence, which can severely degrade the interpretive or recitative performance of those individuals who suffer from them [3]. According to the World Health Organization [4], it can have significant effects on the mental health of the person suffering from it [5].

The major issue with MPA arises when the psychophysiological activation levels of anxiety exceed the individual tolerance threshold, which often results in a significant decrease in their technical abilities [6]. These circumstances are catastrophic in musical performance, which, due to its particular professional characteristics (live public exposure, attention grabbing, impossibility of pause or restart once started, etc.), may well invalidate the professional performance of those who suffer from it, often suddenly [7,8].

The prevalence of MPA among training or professional musicians is alarmingly high. Dalia [9] indicates that between 60 and 80% of professional musicians suffer from its debilitating forms. Additional data on the subject are equally worrying. Lledó-Valor [10] noted in their research that 95% of participants experienced stage anxiety during live performances. In their study involving 570 musicians and 60 teachers aged between 10 and 54, Ballester [11] found that 1 in 3 musicians suffers from MPA. Moreover, 20% of students who choose to leave their musical careers do so because of this issue [6,11].

The professional career of a musician involves the development and training of highly complex motor and cognitive skills over the years, as well as a significant development of artistic sensitivity and understanding [12,13]. The culmination of this journey is performance in front of an audience [10,14].

The high levels of MPA among musicians, both professionals and those in training, often start in the academic centers where music students professionalize. The main conflict is detected in the classroom when the teacher carries out his/her work without considering the dual physical and psychic dimension of his/her student [15]. Students are often expected to uphold high levels of perfectionism, engage in excessive rehearsal, and utilize recordings as models of perfection [16], without attending to their emotional needs or preparing them for the stage.

In Western societies, the traditional educational system of classical music has been consolidating for decades. The inclusion of other styles in their programs is very recent. Young et al. [17] collected a series of characteristics that define this pedagogical system as an extraordinarily demanding sociocultural model based on talent, in which the instrument teacher often serves as the main judge of that talent, and the student is passive, in that he/she does not make interpretive decisions and limits himself/herself to imitate [18].

Unlike pop or jazz music, which often have improvised structures and greater interpretive freedom, classical music demands maximum precision, which both the performer and the audience scrutinize meticulously [19]. This adds significant social pressure to public performances [16]. Given the extremely high demands and pressures that exist in the professional world of classical music, it is no surprise that it is the musical genre with the highest incidence of MPA among its performers [20].

To understand anxiety disorders and their unique nature, Lang’s three-dimensional model has been widely accepted and used in the field of research [21,22]. According to this model, the reactions to threat perceived by the nervous system are categorized into a three-response system: cognitive (in the case of MPA, examples would be reasoning incoherently, negative automatic thoughts about performance or self-concept, or memory errors), physiological (for example, in MPA, hand tremors, agitated breathing, tachycardia, excessive sweating, and dry mouth would be common), and behavioral (primarily in MPA, the attempt to escape the situation or avoidance behaviors) [20,22].

Based on his own research, Barlow [23] proposes a triple vulnerability model: biological, general psychological, or specific psychological linked to specific stimuli [3,24]. Later, Kenny [8] readjusts this model to fit the context of musicians and redefines three more specific factors: early relationships, individual psychological vulnerability, and personal concerns regarding a performance [3,24]. For Kenny [2], MPA is a type of chronic anxiety with acute episodes, the genesis of which can be found in the aforementioned or established biological and psychological individual differences or through specific learning with anxious conditioning. It can be exclusive to situations of musical exposure or coexist with other anxiety disorders, especially with social anxiety. According to this author [8], the anxious reaction can be increased by the action of different variables involved, such as ego threat, fear of failure, or situations of exposure to evaluation. MPA is present throughout the musician’s life and does not necessarily have a direct correlation with the years and hours invested in musical preparation, while it may or may not negatively influence musical performance [7,8].

However, the influence of the social component and the educational system on the development of debilitating forms of MPA cannot be overlooked, being a multifactorial construct whose origins should not fall solely on the individual [15,16,18]. Given the large number of aspects that determine the characteristics of any anxiety disorder, it is considered that they are multidimensional and transactional [25], as the level of activation varies depending on the presence and correlation of various elements, such as the musician’s sensitivity to suffer anxiety at the time of musical performance, the efficiency of the performer, the peculiarities of the environment in which to play, the presence and type of audience, etc. [20,26,27].

According to Barbeau [28], the four dimensions involved in the suffering of MPA are: the cognitive dimension, the affective–emotional dimension, the physiological dimension, and the behavioral dimension. As for its treatment, the techniques on which most of the coping strategies for this condition are based include: the Alexander technique, Feldenkrais technique, applied relaxation by Öst, the Kovács method, systematic desensitization, attention and concentration exercises, anxiety timing, control of negative thoughts, awareness of cognitive distortions, cognitive behavioral techniques, imagined or visualization exposure, strengthening of self-esteem and self-concept, self-suggestion, live rehearsal or exposure, breathing exercises, goal setting, autogenic training, stimulus association technique, self-instruction training (individuals guide themselves through a process of instruction), and rational emotive therapy [29]. On the other hand, following Zhukov [30], the treatment of MPA usually includes mindfulness-based approaches, physiological/physical therapies, cognitive behavioral therapies, prescribed medication, music therapy, and psychotherapy.

According to the research of Herrera, Manjón, and Quiles [31], during a music performance cognitive symptoms predominate to a greater extent than physical symptoms, which makes it necessary, therefore, to develop and implement a treatment program suitable for these characteristics by clinical professionals who can help in coping with this condition and give it the attention it deserves [32,33]. In this sense, Zhukov [30] asserts that the combination of behavioral techniques with cognitive therapy strategies seems to be the most promising approach among interventions aimed at reducing MPA and improving the quality of musical performance. One of the reasons that could explain this fact is that these are more traditional and well-established interventions for different anxiety disorders and are, therefore, better researched. They are also more easily measurable than others.

Nevertheless, the most commonly used coping strategies by musicians to alleviate the negative effects of MPA, both in classical and popular traditions, are alcohol, antidepressants, anxiety medication, beta-blockers, deep breathing, distraction methods, talking about it with family and/or friends, discussing it with the music teacher, consulting a doctor/psychologist/psychiatrist, familiarizing oneself with the venue for the performance, hypnosis, increasing practice hours, simulated performance practice, use of non-prescribed medication, positive self-talk, and relaxation techniques [32]. Of these techniques, those focused on emotions, such as deep breathing, increased practice, familiarizing oneself with the performance venue, and relaxation exercises are predominantly used by musicians suffering from MPA. In contrast, problem-focused strategies, which include seeking professionals and health care services (i.e., external resources), are used less frequently, although they are perceived as more effective [32].

Currently, as Zhurkov [30] recommends, it is necessary to interpret the efficacy of MPA interventions with caution, as many of them have significant methodological and sample shortcomings that prevent them from generalizing to other populations, and furthermore, to establish, with the necessary scientific guarantees, general procedures or action protocols.

Progress in this scientific area has been hindered by a scarcity of publications utilizing methodologies that are both valid and reliable [30]. Therefore, it is advisable to comprehensively accumulate information to significantly enhance our comprehension of this subject. With an absence of studies amalgamating the scientific advancements achieved over the last half decade, the principal objective of this article is to meticulously examine the intervention strategies addressing the issue of MPA implemented from 2018 to 2023.

## 2. Materials and Methods

### 2.1. Search Method 

This review has been conducted following the guidelines established by the preferred reporting items for systematic reviews and meta-analyses (PRISMA) [34,35,36]. Searches were carried out in the Web of Science, Scopus, and Google Scholar databases. The following combined keywords were tracked in English and Spanish: music performance anxiety, intervention, coping skills, regulation skills, coping strategies, coping methods, treatment, psychotherapy, psychological program, intervention program, management strategies, reducing stage anxiety in a performance situation, musical performance anxiety, coping strategies, regulation strategies, treatment, therapy, psychotherapy, coping method, emotional treatment, intervention program, and reduction.

The search strategy employed a combination of Boolean operators and relevant symbols to enhance the retrieval of pertinent articles within the databases. This approach was tailored to the specifications of each consulted database and comprised the following term combinations: (“music performance anxiety” OR “musical performance anxiety” OR “stage anxiety”) AND (“intervention” OR “treatment” OR “psychotherapy” OR “psychological program” OR “intervention program”) AND (“coping skills” OR “regulation skills” OR “coping strategies” OR “coping methods” OR “management strategies” OR “reduction”) AND (“emotional treatment” OR “therapy” OR “emotional intervention” OR “coping method”) AND (“regulation strategies” OR “emotional regulation” OR “coping strategies” OR “intervention program” OR “reduction”).

Below are the criteria used for the process of inclusion and exclusion of the resources reviewed.

### 2.2. Inclusion Criteria

To include the reviewed articles, the following criteria were used: (a) articles present in bibliographic databases of recognized quality, in this case, Web of Science and Scopus, (b) articles published between the years 2018 and 2023, (c) studies that review or investigate treatment methods of MPA in professional or training musicians of both sexes and without age limit, (d) articles and studies in Spanish and English, (e) articles with access to the full study, and (f) experimental and quasi-experimental studies.

### 2.3. Exclusion Criteria

The following exclusion criteria were applied: (a) articles, theses, chapters, and journals on general aspects of MPA, (b) bibliographic reviews, (c) studies to test the validity of certain evaluation tools, (d) academic resources that deal with specific aspects of MPA, whether physiological, psychological, instrumental, or gender, that do not include intervention proposals for their treatment, and (e) research about other types of non-musical performance anxiety.

## 3. Results

In the initial literature exploration, a total of 71 references were identified. After applying the inclusion and exclusion criteria, 61 were discarded, leaving 10 that were extensively analyzed. Figure 1 provides a schematic representation of the selection process for the chosen articles, detailing the aforementioned inclusion and exclusion criteria, in accordance with the PRISMA method:

Upon selecting the 10 studies that satisfied all of the inclusion criteria, we undertook a comprehensive analysis of each. Below, Table 1 summarizes the information about the selected publications, location, design, data collection, type of treatment, control group intervention, duration of the interventions, samples, applied statistical contrast tests and a summary of results.

The countries where the studies included in the review were carried out are, in descending order of predominance: Spain (*n* = 2), United States (*n* = 2), United Kingdom (*n* = 2), Australia (*n* = 1), Austria (*n* = 1), Brazil (*n* = 1), and Mexico (*n* = 1). The intervention modalities used in these studies were, also in descending order: mindfulness and other mind–body strategies (*n* = 2); combination of different types of psychotherapies with mind–body strategies (*n* = 2), specifically, one study used psychotherapy together with visualization, body awareness, breathing and relaxation, and another made use of cognitive behavior therapy together with mindfulness, emotion regulation therapy, optimal experience states and positive psychology jointly; group acceptance and commitment therapy (*n* = 1); acceptance and commitment coaching (a variation of acceptance and commitment therapy) (*n* = 1); compassionate mindfulness (*n* = 1); use of oxytocin (*n* = 1); expressive writing (*n* = 1); and use of pre-concert routines (PPR for its acronym in English). Table 1 shows the reviewed studies, location, design, sample intervention, measurement, and outcomes of interest.

All the evaluated studies are, according to the inclusion criteria, at least quasi-experimental studies. Five of them are uncontrolled clinical trials with a quasi-experimental design of pre- and post-intervention evaluation in a single experimental group. One of them applied a crossover experimental design applying the double-blind technique to the same subjects. Finally, four of them employed a mixed experimental design, including between-subjects and within-subjects measures, and control group. The modalities used for the control group were placebo (*n* = 1), equivalent neutral treatment (*n* = 2), active comparator (*n* = 1), and untreated control group (*n* = 1). Across all studies, sample sizes varied, ranging from a minimum of one participant to a maximum of sixty-two.

Six of the reviewed studies included both sexes in their samples, two included only male participants and two did not specify the gender of their participants. The subjects that made up the samples have been mostly adults (over 18 years), focusing almost entirely on young adults. Thus, seven of these investigations present mean ages that range between 18 and 30, while only one study presents an age range between 40 and 50. The remaining study does not specify the age of its participants (*n* = 1). None of the samples included minors among its participants.

In terms of musical background, five of the studies included students from music universities, one consisted of students from pre-college music programs, and four made up their samples of participants with varying musical developments, be they students and professionals (*n* = 2), students and teachers (*n* = 1), or students, professionals, and enthusiasts (*n* = 1). No study evaluated a sample composed entirely of professionals.

Six studies featured participants from a single specialization. These are, in descending order: voice (*n* = 3), choir voice (*n* = 1), piano (*n* = 1), and violin (*n* = 1). In the remaining four cases, the samples were made up of musicians from different instrumental specialties, including voice, piano, and violin along with other mixed specialties (unspecified woodwind and brass instruments, viola, cello, guitar, bass, jazz double bass, jazz saxophone, and electronic music instruments or devices).

Despite a thorough reading of the research suggesting that some of these are based on samples with participants trained in classical music, most of them do not make this specification clear. Thus, we find that the detailed musical styles are: classical music (*n* = 1), contemporary commercial music (*n* = 1), musical theater (*n* = 1), and mixed samples of various styles, such as classical music, jazz, and electronic music (*n* = 1), or classical music and other (non-specific) styles (*n* = 1). The other five studies do not clarify the musical style in which the participants of their samples have been trained.

In almost all the studies, the level of MPA manifested by the subjects was taken into account, although it does not specify when the normal adaptive levels were exceeded in intensity and persistence and when they were within regularity according to the pre-test measures in the positive cases.

In the five studies that employed an inter-group experimental design, it was observed that the control groups were similar to the corresponding experimental group in sociodemographic terms, and the participants of the control group were recruited from the same places as the experimental groups.

Eight studies based their conclusions on the reduction in psychological indicators (psychological flexibility and vulnerability, distancing from specific thoughts, sensation of control, self-awareness, acceptance of MPA-related discomfort or changes in perceptions of specific elements of the stressful situation), while two of them combined psychological and physical data (cardiac and salivary evaluation).

The psychometric questionnaires used are diverse and different in each study, although it is worth highlighting the use of the K-MPAI (either in its original version or in its Spanish translation). In this review, it is used along with other measurement tools by six out of the ten evaluated studies. The rest of the standardized questionnaires were used only in a single study, with no repetition of any other tool in any of them.

The results indicate positive outcomes in measured psychological and physical variables for eight of the ten cases. In contrast, two showed no significant changes. In these two cases, the methodologies implemented were (1) compassionate mindfulness, with measurements taken via pre- and post-test behavioral, cardiac, and salivary psychometric evaluations, and (2) mindfulness, yoga, and other mind–body exercises, where cortisol levels were gauged using pre- and post-test saliva samples. For studies yielding positive conclusions, methods did not include physiological measurements, save for one that measured heart rate but did not evaluate cortisol levels in saliva. This suggests that studies incorporating salivary samples for cortisol level determination yielded negative outcomes. It is noteworthy that similar intervention strategies, such as mindfulness or other mind–body strategies, were employed in studies with both favorable and unfavorable results.

## 4. Discussion

MPA and its impact on musicians’ skills during performances is an emerging area in the scientific literature, with increasing research into mitigating interventions [7,30,33]. The recent literature has provided a more detailed understanding of MPA than previous research [30,33,59]. While theoretical advances and treatment proposals exist, empirical research evaluating the efficacy of these treatments is limited, and even scarcer when considering studies with a randomized and controlled design. In addition, sample sizes are small, which limits the interpretation and generalization of the results. These general conclusions are consistent with those made several years earlier by Zhurkov [30], who agreed in the analysis of the methodological and sample shortcomings that prevent generalization.

More specifically, according to the reviewed research, no significant differences are observed in the improvement of MPA when differentiated by musical instruments. Additionally, in cases where results have been differentiated by gender, it is concluded, in a manner consistent with other studies [6,31,60,61], that women experience a higher level of MPA compared to men.

The examined studies demonstrate higher methodological robustness compared to the studies included in previous reviews [2,7,33,59]. However, they still face challenges in their experimental designs while striving for greater consistency and precision in the results. Therefore, a cautious interpretation is necessary for much of the evidence presented in this study since it does not stem from randomized inter-group experimental studies with large population samples [30].

It is important to note that the majority of the studies do not specify the levels of MPA experienced by the participants, particularly when they reach debilitating degrees of the condition. This information is crucial for drawing conclusions about the effectiveness of the evaluated treatments. Furthermore, many of these investigations have other limitations, such as a limited number of participants studied, predominantly composed of young adults and music students, a lack of standardization of tools and treatment duration, and occasionally the use of non-validated outcome assessment instruments [62,63].

There is another relevant issue that must be taken into account, and that is the diversity of tools used to measure the same variables. The comparison of results would be more straightforward and precise if there were standardization of the instruments used, as well as a more homogeneous delimitation of variables to be measured, as also pointed out in previous study reviews [33]. In this regard, the use of the K-MPAI questionnaire [8] is common, being the most widespread and utilized method for measuring MPA (used in seven out of the ten cases).

The intervention proposals used in the reviewed studies encompass a variety of approaches, ranging from more traditional or established ones in our mental health system (psychotherapy, cognitive behavioral therapy, acceptance and commitment therapy, and their variants) [37,41] to treatments that include exercises from Eastern philosophies and mind–body training worked on in a mixed manner, alongside psychological therapies [24,40] or in isolation [3,10,38,42,64]. It is, in fact, remarkable that five out of the ten studies predominantly or exclusively include proposals of this second type in their interventions, suggesting they are gaining prominence.

The contrast between studies may be of little significance due to the lack of homogeneity among their designs. The variability in how MPA is measured in different studies complicates the synthesis of results.

It is noteworthy that none of the studies includes subjects under the age of 18, which is associated with a lack of early attention to this condition, which worsens over time. Studies on social anxiety and performance anxiety (of which MPA is a specific manifestation) reveal a bias in this age group linked to the absence of early monitoring, diagnosis, and intervention, as it is considered something common among musicians [33,65]. Thus, the perception of a lack of attention and intervention for MPA is greater in contexts where its presence is seen as insignificant and inherent to the profession and/or musical training [33]. This fact implies a potential risk that these emerging musicians may abandon their musical careers if the levels of MPA become exceptionally intense, surpassing normal adaptive responses.

Another aspect concerning the treated populations is the absence of interventions solely focused on professional musicians and those above 40 years of age (in this case, only one study meets this condition). It would be interesting to include participants with these characteristics since, in many cases, they end up resorting to coping strategies involving the use of drugs, alcohol, or medication without medical supervision [45,66,67].

Initial music education institutions require a rigorous review of their pedagogical methodology due to its impact on the manifestation of MPA (Belmonte, 2018). Taking this into account and based on this review, it can be concluded that the treatment of this condition in students and professionals who already suffer from it remains an unresolved issue. This begins at early stages with a prompt detection of cases of debilitating forms of MPA, something that, as it has already been mentioned, is often neglected due to the belief that it is a normal part of the musician’s profession [33,65]. In addition to this initial assessment, it is necessary to implement a standardized evaluation of the degree of MPA and, based on this pattern, appropriate intervention. These stages must be reviewed, standardized, and accorded the importance they deserve, given the concerning state of mental health among professionals in the field, as well as aspiring musicians who undergo training in environments and pedagogies that largely ignore emotional management in this regard [15,18].

Therefore, it is crucial for younger students to be aware of the depth of this condition, recognize it, seek help, and even have tools to manage it. The alarming normalization of drugs use to alleviate MPA symptoms among musicians of all ages should be considered [68] and replaced with approaches that address the underlying components of this condition, not just its physiological symptoms.

In the coming years, it would be advantageous to work with techniques that have proven to be more effective in their results, such as cognitive behavioral therapy, used in isolation or combined with other strategies like music therapy or mindfulness [68], and standardize data collection tools to establish a protocol for addressing this situation.

## 5. Limitations and Strengths

Heterogeneity in the research designs and instruments employed may impact the coherence and interpretability of the findings within this review. In the ten selected studies, several limitations were observed. Five out of the ten studies suffered from a small sample size, lacked a control group, and solely utilized cognitive assessments. Within these five, four did not have a post-intervention follow up, and one implemented an intervention without prior validation. Among the inter-group studies (*n* = 5), three were characterized by a small sample size and the absence of a post-intervention follow up. One of these studies used a crossover design, with the same participants serving in both the experimental and control conditions at different times. This study applied a physiological intervention but limited its assessment to cognitive measures and neglected the post-intervention follow up. Additionally, another study in this category had a control group that received no intervention, blurring the lines between whether observed outcomes in the experimental group were attributable to the treatment or merely a consequence of experimental maturation.

This review may present other limitations, starting with the identification phase, as it is possible that not all relevant studies were found, especially if they were published in less well-known journals or in languages different from English, Spanish, or Portuguese. Furthermore, it is not possible to guarantee complete objectivity in the selection and evaluation of the articles, as decisions on which studies to include and how to assess them may be influenced by subjective judgments. Additionally, given the rapid evolution of research in psychology, a review may quickly become outdated. Finally, the lack of homogeneity among the designs may render the contrast between studies less significant, as previously mentioned.

Regarding its strengths, this review has identified patterns and weaknesses in the designs, which allows for establishing the effectiveness of the reviewed interventions and may influence decision making and the direction of future studies and research.

## Figures and Tables

**Figure 1 behavsci-13-00720-f001:**
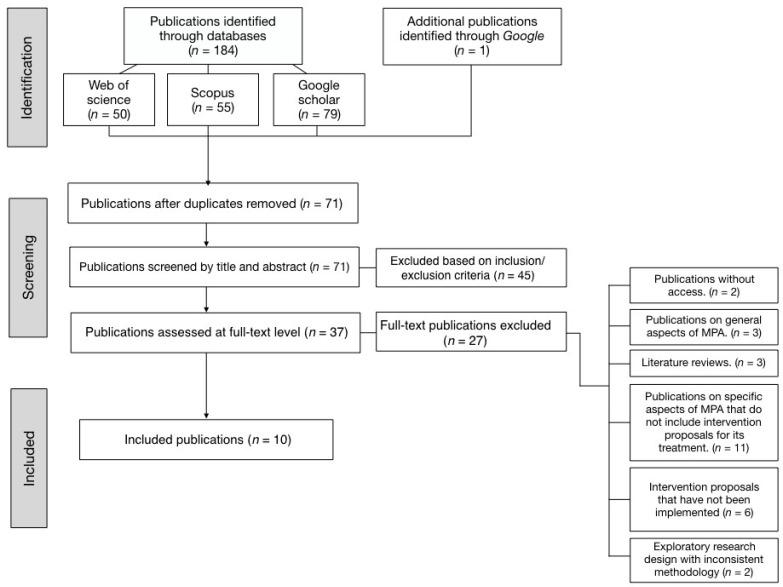
PRISMA screening process.

**Table 1 behavsci-13-00720-t001:** Selected publications, location, design, data collection, type of intervention, control group intervention, duration of intervention, summary of the most pertinent results, samples, and applied statistical contrast tests.

Publications	Design	Data Collection	Type of Treatment	Control	Duration of Intervention	N (GE/GC) and Participant Profiles	Statistical Contrast Tests Applied	Summary of Results
Clarke, O. & Baranoff (2020) [37] (Australia)	Uncontrolled trial. Quasi-experimental within-subject design with pre- and post-treatment measurements. Follow up over three months.	K-MPAI, MPFI-SF DASS, and MHC-SF. Open questions	Group therapy of acceptance and commitment	No control group	Six weeks. One 2 h session per week.	6 (6/-) University music students in their 1st, 2nd, or 3rd year of studies	Repeated measures analyses of variance (ANOVA)	A significant and sustained reduction in MPA was observed over time (Huynh–Feldt correction F (1.11, 5.58) = 9.48, *p* = 0.019, partial η 2 = 0.655).
Cortés-Hernández et al. (2023) [3] (Mexico)	Uncontrolled trial. Quasi-experimental within-subject design with pre- and post-treatment measurements, without follow up.	IDARE, MAAS, and ECOM	Mindfulness and compassion therapy	No control group	Eight weeks. One 2 h session per week.	20 (20/-) University music students	McNemar’s Test and the Wilcoxon Rank-Sum Test	No significant changes in trait and state anxiety or physiological variables.
Czajkowski et al. (2022) [38] (United Kingdom)	Uncontrolled trial. Quasi-experimental within-subject design, without follow up.	FFMQ, ad hoc questionnaire MFM, and semi-structured interview	Mindfulness through the MfPAS course	No control group	Eight weeks. Weekly group session of 2–2.5 h. Daily individual practice (Routine) of 40–45 min.	25 (25/-) University music students	Paired *t*-tests and related non-parametric samples Wilcoxon	Indirectly reported improved MPA via Mindfulness tests (*p* < 0.01). Qualitative data showed varied focus on positive effects on MPA symptoms.
De Lima-Osorio et al. (2022) [39] (Brazil)	Crossover experimental design applying the double-blind technique to the same subjects.	K-MPAI, SSPS-P, and SRQ-20	Oxytocin (pharmacological treatment)	Placebo	Two measurement sessions with a distance of 15 to 30 days.	50 (22/28) Music students and professionals	ANOVA 2 × 2 for crossover trials	Oxytocin improved the cognitive component of MPA (*p* = 0.06).
Fernández- Granados y Bonastre (2021) [24] (Spain)	Uncontrolled trial. Quasi-experimental within-subject design with pre- and post-treatment, without follow up.	Open questions	Psychotherapy, visualization, body awareness, breathing, and relaxation	No control group	Eight sessions over two weeks (Each session lasts 15 min.)	46 (46/-) Amateurs, beginners, pre-college or university students, and professionals	Analysis of variance (ANOVA) and Student’s *t*-tests	The results were not statistically significant in the Psychological Vulnerability, Specific Thoughts, Motor and Physiological scales overall, but were significant in some items of each scale.
Moral-Bofill et. al. (2022) [40] (Spain)	Controlled trial. Mixed experimental design, between-subjects and within-subjects, with pre- and post-treatment measurements, without follow up.	EFIM, KMPAI-E, and SSS	HAMI method. Cognitive behavioral therapy (CBT) exercises, mindfulness, emotional regulation therapy, optimal experience states, and positive psychology	Untreated control group	Not specified.	62 (28/34) Students in their final year of university and university professors	Structural equation modeling (SEM), repeated measures mixed ANOVA, and Student’s *t*-test	Intervention significantly decreased MPA (t = 2.64, *p* = 0.01, d = 0.24), and self-awareness (t = −3.66, *p* = 0.00, d = 0.70) in the experimental group.
Shaw et al. (2020) [41] (United Kingdom)	Uncontrolled trial. Single- subject design. Measurements taken before, at the midpoint, and after the intervention. Follow up after three months.	BAFT, AAQ-2, PHLMS, and K-MPAI	(ACT) acceptance and commitment coaching (version of commitment therapy)	No control group	One introductory session and six 1 h sessions over four months.	1 (1/-) University musical theater student and professional	Reliable change index	Clinically significant improvements in two ACT-based processes believed to correlate with improved psychological flexibility in previous ACT for MPA psychotherapy research, i.e., acceptance of MPA-related discomfort and defusion from MPA-related thoughts.
Shorey (2020) [42] (United States)	Controlled trial. Mixed experimental design, between-subjects and within-subjects, with pre- and post-treatment measurements without follow up.	K-MPAI and FZA-S	Mindfulness, yoga, and other mind–body exercises	Equivalent neutral treatment	A single session.	16 (8/8) University music students	One-tailed *t*-test and Regression model	There were no significant improvements in cortisol reactivity or state MPA.
Tang & Ryan (2020) [43] (United States)	Controlled trial. Mixed experimental design, between-subjects and within-subjects, with pre- and post-treatment measurements, without follow up.	QSMEEPH and expressive writing	Expressive writing	Equivalent neutral treatment	Two sessions two to three days apart.	35 (18/17) University music students	Mixed-design ANOVA	Students with high levels of MPA significantly reduced it (F = 4.99, *p* < 0.04).
Tief & Gröpel (2021) [44] (Austria)	Controlled trial. Mixed experimental design, between-subjects and within-subjects, with pre- and post-treatment measurements, without follow up.	K-MPAI, MRF-3, and SMPQ	Pre-concert routines (goal setting system = control group)	Active comparator (a goal setting system applied to control group)	Two sessions 33 days apart.	30 (15/15) Music students and professionals	2 × 2 repeated measures ANOVA and chi-square test	There were no significant differences observed in MPA in either of the two intervention techniques.

Meaning of the acronyms that appear in Table 1: AAQ-2 = The Acceptance and Action Questionnaire-2 [45], MPA = Music Performance Anxiety (when it appears in Measured Variables it includes three subcategories: concerns and insecurity, depression and hopelessness, and early parental relationship), BAFT = The Believability in Anxious Feelings and Thoughts [46], CBT = Cognitive Behavior Therapy, DASS = Depression, Anxiety, and Stress Scale [47], ECOM = Compassion Scale for Mexican Population [48], EFIM = Spanish acronym for Flow State Scale for Musical Performers [40], EW = expressive writing, FFMQ = Five Facet Mindfulness Questionnaire, FZA-S = Questionnaire for Appearance—Self-assessment (English translation [49]), HAMI = Spanish acronym for Self-Regulation Skills for Performing Musicians, IDARE = Trait-State Anxiety Inventory [50], K-MPAI = Kenny Music Performance Anxiety Inventory [8], KMPAI-E = Spanish version of Kenny Music Performance Anxiety Inventory [8], MAAS = Mindful Attention and Awareness Scale [51], MFM = Mindfulness for musicians, MfPAS = Mindfulness for Performing Arts Students course, MHC-SF = Mental Health Continuum-Short Form [52], MPFI-SF = Multidimensional Psychological Flexibility Inventory-Short Form [53], MRF-3 = Mental Readiness Form-3 [54], PHLMS = The Philadelphia Mindfulness Scale [55], PPR = Pre-performance routines, QSMEEPH = questionnaire of students’ music educational experience and performance habits [2], SMPQ = Self-Efficacy for Musical Performing Questionnaire [56], SSPS-P = Self Statements During Public Performance, SRQ-20 = Self-Reporting Questionnaire [57], SPST = Simulated Public Singing Test, SSS = Social Skill Scale [58].

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
