# Peer review of "Current Trends in Music Performance Anxiety Intervention"

_behavsci, 2023, doi:10.3390/bs13090720_

Round 1

Reviewer 1 Report

Current trends in music performance anxiety prevention and intervention

1.     The discussion is compelling as it provides valuable insights into music performance anxiety (MPA) among musicians at different stages of their training. It highlights the growing attention to this issue in the scientific literature and the increasing focus on interventions to alleviate its harmful effects. The discussion touches upon various important research findings, limitations, and potential future directions.

2.     The article mentions music performance anxiety as a "cognitive disorder." While anxiety can have cognitive components, it is more accurately described as an emotional and physiological response to stress. Clarifying this aspect would ensure accuracy in the description.

3.     Expanding on the potential implications of the study's findings may include how these advances in preventing and treating music performance anxiety impact the well-being and career success of professional musicians and musicians in training.

4.     It might also be beneficial to include specific gaps that need to be addressed or potential new approaches that could further advance the understanding and management of music performance anxiety.

 The article has some misspelled words, wrong punctuation, grammar lapses and dangling modifiers (like in the Abstract).  Consider subjecting the article to Grammarly or other grammar check devices.  Some sentences were stated vaguely. This article would benefit from close editing and proofreading.

Author Response

Thank you very much for your positive feedback on our manuscript. We have improved each section to provide a more comprehensive background and 10 new relevant references, better methodological consistency, and to address all doubts and suggestions raised by the reviewers.

We greatly appreciate these suggestions. In total, the changes and improvements implemented have involved more than 2000 additional words, in spite of the exclusion of two of the studies and 5 tables included in the initial version of the article. A new Figure 1 detailing the PRISMA screening process, the consolidation of all tables into a single Table 1 incorporating new relevant data, the inclusion of a new Table 2 with a summary of results, a new section on study limitations and strengths, and the rephrasing or omission of certain parts of the text in order to make the provided information more consistent.

Below, we provide a point-by-point summary of our response to your comments. These improvements are clearly highlighted in red in the attached revised manuscript. These modifications have substantially enhanced our work, and we are very grateful for your valuable input.

Reviewer 2 Report

This is a good review idea which could potentially shed a new light on the most recent literature around MPA. However, findings are very unclear, and the review’s contribution to the literature is not known. Furthermore, there is confusion between a literature review and a systematic literature review, the presentation of results could be greatly improved, and overall, the writing could benefit from the help of a native English speaker (especially the discussion section). However, the lack of clarity around the contribution of this review alongside the lack of clarity around the results are the greatest weaknesses of this work. Please find below more specific comments which could help improve this work:

Abstract:

In the abstract you make some statements that are untrue or highly debatable. You call MPA a disorder, which is problematic. MPA, to some extent at least, is a normal reaction to one’s environment (i.e. the presence of a public, the perception of high stakes, the pressure to conform to certain norms, the focus on not making mistakes, etc.). I would avoid pathologizing a normal reaction to such an environment. If you are referring to debilitating forms of MPA, rather than MPA, then please be explicit. Also, although in the introduction section you talk about non-cognitive aspects of MPA (such as physiology and behaviour), in the first sentence of the abstract you call MPA “cognitive”. I suggest that you take this out, as it seems to be making the individual responsible for a phenomenon that has much wider causes (including social, ideological, etc).

You also mention that it is “the greatest perceived threat to their professional work in this population” – well, who decides if it’s greater or as great as performance-related musculoskeletal injuries, for instance? More cautious language would be appreciated. Agreed that MPA may be one of the greatest perceived threats to their professional work, but whether it’s the greatest, I’m not sure…

Introduction:

Again, this is over-pathologising MPA. Also, calling it a phobia is problematic – the evidence you provide is unconvincing. Also, there is way too much focus on the individual here and making the individual responsible (by calling it a pathology), without questioning the norms and pressures in Western classical music, which maybe contributing to MPA (see Leech-Wilkinson, 2020: https://challengingperformance.com/the-book-14/).

A more critical approach to Kenny’s work would have been welcomed, since it focuses almost exclusively on the individual. You yourselves point that out in the next paragraph: “Given the many aspects that determine the characteristics of any given anxiety disorder, anxiety disorders are considered to be multidimensional and transactional”

A more critical discussion of the literature in general would be welcomed. For instance, you make some good points here: “Currently, as recommended by Zhurkov (2019), it is necessary to interpret the effectiveness of interventions in MPA with caution since many of them have important methodological and sample shortcomings that prevent them from being generalized to other populations; or established with due scientific guarantees, general procedures or protocols for action.” I would also add that one of the reasons why cognitive therapy with behavioural elements may be “the most promising approach” could be the fact that this therapy is easier to measure, more popular than others, and more often researched. After all, why wouldn’t talking to an empathetic friend who listens well be equally effective for mild forms of MPA?

Last sentence – could you please shorten and simplify this? “In this sense, this theoretical review sets out, as its main objective, to review the various intervention strategies carried out to address MPA, whether preventive or treatment, in the last five years (2018-2023), analyzing the particularities and techniques associated with each treatment approach in scientific studies published in impact journals that review or investigate methods of treating MPA in professional musicians or musicians in training”. Also, not sure what is meant here by “impact journals” – do you mean peer-reviewed journals? If yes, please clarify.

Materials and Methods

Inclusion criteria: please add a comma before c), instead of a full stop.

Exclusion criteria: Please amend this: “The following criteria were used to exclude excluded material”. Also, please add the necessary commas between points a), b), c), d), and e). Re d), I think you mean “that DO not” instead of “that does not”.

“As a result of this process, a total of 71 references were identified, of which 59 were discarded, and 12 were selected and worked on exhaustively.” – this should go into the results section, not here.

Re Figure 1, I was expecting to see a PRISMA flowchart which includes more information about how many abstracts you’ve included, and then which papers made it to the full-text stage. Also, “preselected publications” sounds a bit odd. Have you also conducted a manual search of the references lists of the included papers? Finally, it would be interesting to see how many papers you’ve retrieved after having conducted the search strategy on each of the databases you mentioned, and how many duplicates you’ve removed (see the PRISMA flowchart template). It still is unclear to me if this is a systematic literature review or a literature review. Under “Search method”, you call it “systematic”. However, if so, then I’m afraid you’re missing the quality appraisal tool which is central to a systematic literature review.

Results

“the samples varied in number of participants” – please take out “in number or participants” as this is redundant.

Table 1 – I wouldn’t include the titles of the included papers, as this doesn’t add much. However, I would add a brief summary of their results (instead of having these separately in table 7!), and the control groups they used (i.e. waitlist, no treatment, etc.). As for the type of treatment column, please remove all the full stops, as these aren’t complete sentences. In terms of results, please provide actual numbers and p values, since you’re looking at experimental designs.

“Regarding gender, seven of the studies reviewed included both sexes” – please don’t use gender and sex interchangeably. Sex is biological, gender isn’t. Please clarify if you meant gender or sex. Note that you also mention gender in the inclusion criteria, so please clarify.

“while only one study has a mean age between 40 and 50 years.” – this is a range, not a mean.

“Only one of the samples has children and adolescents among its participants, namely subjects between 10 and 20 years of age” – OK, but earlier you seemed to be referring to adults as “18 years of age”, and now you’re referring to them as adolescents. Please be consistent.

“Table 2 below shows the number, gender, and age of participants in the Experimental and Control Groups” – no need for capitalization here!

Table 2 – I would remove this completely – it’s unnecessarily detailed and I’m not sure what it adds. You can easily summarise information on gender/sex, age, and sample size across all included studies, in a more narrative form.

“As for the origin of the samples” – please change this to “In terms of musical background”

Not sure what you mean by “regulated music colleges” – are there un-regulated music colleges? Also, what is meant by intermediate students? Please specify. I don’t know what is meant by “higher grade students” in Table 3. I think Table 3 can easily be summed up in a narrative format – I don’t think we need to know that level of detail. After all, you need to summarise findings ACROSS studies, and not for each study separately. The same applies to Table 4 – I find it’s unnecessary, since you’ve already summarized the relevant findings.

“In almost all studies, the level of MPA manifested by the subjects was taken into account, although it is not indicated when it was considered a pathological level or within normality according to pretest measures in positive cases” – overly medical language for something that is so complex and potentially non-medical. Please amend.

Some elements of Table 5 such as the characteristics of the interventions could easily be incorporated into table 1, especially if you take out the titles column from table 1. As for the data collection column, this is unnecessary. Also, take out the qualitative bits, as you’re not interested in these, even if the authors of the original papers included qualitative data. You should only mention the parts of the articles that are relevant to your inclusion criteria. Again, info about design type should appear in table 1. You don’t need so many tables. One table that includes the essential information and is comprehensive enough should do (and you could include it in landscape format). You might also want to look at published systematic literature reviews to see how others have done it. Furthermore, table 5 (or table 1 for that matter), should clarify the outcomes of interest, and not all the outcomes that have been reported in the original papers. Please clarify what the outcomes of interest are (this should only be MPA – the only thing that you can add is HOW it’s been captured (i.e. by what questionnaire).

Table 6 should in fact appear under table 5, as further notes about the acronyms, not as a separate table (and remove the author and year column – this is not needed at all!)

I’m not sure what you mean by “psychometric questionnaires” – doesn’t really make sense. I assume you mean validated questionnaires as opposed to ad-hoc ones?

“Some studies have as a comparative pre/post-test reference the quality of the participants' interpretation before and after the intervention based on expert evaluative criteria (n=3), which is still a subjective measure. The other nine studies based their findings on decreases in physical (cardiac, salivary assessment) and psychological indicators of MPA (psychological flexibility and vulnerability, distancing from specific thoughts, sense of control, self-awareness, acceptance of MPA-related discomfort, or changes in perceptions of specific elements of the stressful situation). Three of these studies combined physiological and psychological data, while six of them used only psychological measures. These seven studies based their results on a combination of qualitative (semi-structured interviews, self-reports, and focus groups) and quantitative data (questionnaires and physiological variable measures).” – please take out the part around performance quality. This is irrelevant to your review. Also, I’m not sure I understand what you mean here. Do you mean that some papers assessed MPA objectively and others subjectively? If so, then this should have been clarified in one of the tables (alongside the outcomes of interest). Please clarify this paragraph. I’m not sure why you keep talking about qualitative data, since this does not adhere to your inclusion criteria.

“Other measurement tools used are semi-structured interviews, open-ended questions, focus group discussions, and ad hoc questionnaires. Some studies combine these data collection instruments with standardized questionnaires (n=3), and others use them alone (n=2). The remaining research worked with standardized questionnaires (n=7).” – it’s really unclear why you talk about semi-structured interviews, since you don’t seem to have included that kind of data.

Table 7 – this needs to be included in that main table which reports information about authors, location, design, sample intervention, outcomes of interest, measurement, and findings. If findings are separate, that is possible. However, these need to include actual results/numbers, and p values, since most of them (if not all) seem to be quantitative studies. You still report findings on performance quality, which is confusing. You need to clarify this in the inclusion criteria, as I had implied you were only looking at MPA as an outcome of interest. Please clarify your outcomes of interest and provide a justification.

Discussion

“The effect of MPA on the musician population at any stage of their training, and its strong influence on the deterioration of skills during public performances, is a little-explored topic but increasingly frequently addressed in the scientific literature (Fernholz et al., 2019), as well as the theoretical proposal or implementation of interventions that can alleviate its harmful effects (Burin & Osorio, 2016).” – pleas shorten and simplify this.

“After the literature review carried out, it is possible to confirm that there is more and more information and evidence that helps to understand this ailment more specifically than in previous reviews (Brugués, 2011; Burin & Osorio, 2016; Zhurkov, 2019), with the consequent improvement in proposals and tools that help in its treatment at a theoretical level, but there is relatively little research at present that has aimed to evaluate the effectiveness of a treatment for MPA and even less research that has carried out this process in a randomized and controlled design.” – please be less vague/more specific.

“Moreover, the number of subjects in the samples is, in almost all cases, limited, which conditions the interpretation and generalisability of the results.” – change this to “which limits the…”

“greater affection on the part of instrumentalists who were of the classical genre” – please rephrase this, as it sounds awkward. Also, this doesn’t exactly come across from the findings you reported…

“The studies analyzed present greater methodological strengths than those carried out in previous years' reviews (Brugués, 2011; Burin & Osorio, 2016; Fernholz et al., 2019; Kenny, 2005).” – vague. What do you mean?

“the use of descriptive rather than statistical analyses” – surely you mean “the use of descriptive rather than inferential statistics”. However, I’m assuming you’ve only included studies that did some form of inferential statistics!

“strong guarantees of reliability and validity.” – I’d embrace a more cautious language. There are hardly any guarantees in science.

“At this point, it is important to note that most studies lack information on the levels of MPA experienced by participants, especially in cases of pathological levels” – please avoid pathologizing MPA. Are you referring to debilitating forms of MPA? If so, clarify. Otherwise, MPA is not a pathology!

“Studies on social anxiety and performance anxiety (pathologies of which MPA is a part)” – there is no evidence to suggest that MPA is a pathology, and I think it is very problematic to refer to it as such!

“This fact contemplates the possible consequences of many of these budding musicians abandoning their musical careers due to the conditions that accompany suffering from a pathology such as this.” – awkward phrasing. A fact cannot “contemplate”. And again, please change the pathology-related narrative, as this is a very reductionist approach to MPA.

“It would be interesting to deal with participants with these characteristics, as they have been able to create their coping strategies, including the use of medication or even drugs or alcohol.” – please rephrase for clarity, as it’s currently awkwardly phrased.

“The centers where music training begins at an institutional level need a profound review and methodological reflection as they are the epicenter of the onset of MPA” – I don’t understand how the centers could need “methodological reflection”. Also, “the epicenter of the onset of MPA”? I don’t know what you mean, but perhaps some caution would be desirable.

More details around the implications of these findings for future research and practice would have been welcomed. Also, what are the limitations and strengths of your review?

I suggest seeking the help of a native English speaker (especially for the discussion section). Also, more cautious language is advisable, given that this is a piece of academic writing. 

Author Response

(The authors gave the same response as above.)

Round 2

Reviewer 2 Report

Thank you for engaging with the comments and improving the narrative around MPA, the methodology section, table presentation, and the discussion. Please see below some further comments aimed at helping you improve this further:

Introduction:

“However, the influence of the social component and the educational system on the development of debilitating forms of MPA cannot be overlooked, being a multifactorial construct whose origen should not fall solely on the individual” – please correct “origins”

“According to Barbeau [28], the four dimensions involved in the suffering of MPA are: the cognitive dimension, the affective-emotional dimension, the physiological dimension, and the behavioral dimension; and as for its treatment, the techniques on which most of the coping strategies for this pathology are based include” – please don’t call it a “pathology”

“In spite of these advances” in your last paragraph – this seems odd, given that it comes after the paragraph on the methodological weaknesses of current interventions, rather than any advances… Please consider rephrasing.

Materials and Methods:

You mention that “Searches were carried out in the Web of Science, Scopus, and Google Scholar databases from July 2022 to February 2023”, but your abstract and inclusion criteria refer to the last 5 years… please clarify and amend throughout the manuscript for consistency.

Please include an actual search strategy, with the relevant Boolean operators and symbols.

If this is a systematic literature review, then this needs to include a quality appraisal tool as mentioned in my previous review round. The quality appraisal tool allows you to assess the quality of the papers you include and is what distinguishes a systematic review from a literature review. This is also mentioned in the PRISMA guidelines. Also, once you do this, I suggest adding “systematic literature review” in the title and abstract, not just here. If you choose not to include a quality appraisal tool, then I suggest this should be a literature review, and not a systematic literature review. 

Inclusion criteria mention English, Spanish, and Portuguese, but before the inclusion criteria you mention the searches were conducted in English and Spanish only. Please amend.

Table 2 – please keep the results column much more concise, avoid writing in complete sentences, and you need to include p values and effect sizes and/or confidence intervals (wherever applicable). For instance, the results for study no 2 (“No significant changes were observed in levels of trait and state anxiety and physiological variables. However, data shows a statistically significant increase in the level of compassion and a tendency for improvement in mindfulness levels.”) could be rephrased as such: “No sig. changes in trait and state anxiety or physiological variables”. Please don’t report any results that have nothing to do with MPA, such as compassion or mindfulness. Check this throughout. Please consider merging tables 1 and 2 (but you need to keep the results column very concise). Also, the sample size column should also specify who the participants were (professionals, music students, etc.). Finally, consider a landscape orientation only for this table (if merged with table 1).

Table 1: the acronyms in the data collection column need to be explained. Also, make sure these are all measurements of MPA.

Results:

“For studies yielding positive conclusions, methods did not include physiological measurements, save for one that measured heart rate but didn't evaluate cortisol levels in saliva. This suggests that studies incorporating salivary samples for cortisol level determination yielded negative outcomes. It's noteworthy that similar intervention strategies, such as mindfulness or other mind-body strategies, were employed in both positively and negatively concluded studies.” – “positively and negatively concluded studies” sounds really odd. Please rephrase. It is not about the conclusions of these studies, but rather about their results. Also, avoid contractions such as “didn’t” and “It’s” in academic writing.

Re your included studies, which ones were aimed at prevention, and which at treating MPA? I don’t think this has been addressed…

Discussion:

Please use more concise writing. For instance, “In addition, the number of subjects that make up the samples is, in almost all cases, small, which limits” can be changed to “In addition, sample sizes are small, which limits”. Avoid using unnecessary words.

Second paragraph: Is there a reason why here you spell out MPA all of a sudden (i.e. “More specifically, according to the reviewed research, no significant differences are observed in the improvement of MPA (Music Performance Anxiety”)?

“The examined studies demonstrate greater rigor compared to previous reviews” – you mean compared to studies included in previous reviews? If yes, then please clarify. Also, explain what you mean by “greater rigour”, as it’s a little vague…

“The intervention proposals used in the reviewed studies encompass a variety of methodologies, ranging from more traditional or established ones in our mental health system (psychotherapy, Cognitive Behavioral Therapy, Acceptance and Commitment Therapy, and their variants)” – please avoid using the word “methodologies” like this, as it can be confused with research methodologies. Use “approaches” instead.

“This starts from the early stage with an early diagnosis, something that, as we have already mentioned, is often neglected due to the belief that it is a normal part of the musician's profession. In addition to this initial assessment, it is necessary to implement a standardized evaluation of the degree of MPA and, based on this pattern, appropriate intervention.” – Please decide as to whether you consider MPA a normal response (as you claim in your abstract) or a “diagnosis”, and amend accordingly. Also, what evidence is there to suggest that neglecting MPA is due to the belief that it is a normal part of the musicians’ professions?

Limitations and strengths:

Second paragraph: Please refer to the review as “this review” rather than “this study”, to distinguish it from the included studies.

Some minor editing required. 

Author Response

Again, thank you for your corrections and contributions. We fully agree with your criteria and have endeavored to respond to all suggestions, including the suggested format of a single summary table (changes are highlighted in red in the text).
